# Outcomes After Transjugular Intrahepatic Portosystemic Shunt in Cirrhotic Patients 70 Years and Older

**DOI:** 10.3390/jcm9020381

**Published:** 2020-01-31

**Authors:** Natasha Adlakha, Mark W. Russo

**Affiliations:** 1Division of Gastroenterology, Atrium Health-Carolinas Medical Center, Charlotte, NC 28203, USA; 2Division of Hepatology, Atrium Health-Carolinas Medical Center, Charlotte, NC 28203, USA

**Keywords:** liver, encephalopathy, ascites, varices, stent

## Abstract

Transjugular intrahepatic portosystemic shunt (TIPS) is effective at treating ascites and variceal bleeding but may be associated with increased morbidity and mortality in older patients. Our aim was to report outcomes in patients 70 years and older who underwent TIPS because data are limited in this population. We performed a retrospective review of patients who underwent TIPS at our institution over 10 years. We matched those 70 years and older to those 50–59 years old by year of TIPS and the Model for End-Stage Liver Diseae-Sodium (MELD-Na). Thirty-day readmissions were higher in the elderly group (*n* = 50) compared to the younger group (*n* = 50), *n* = 17 (34%) and *n* = 6 (12%) (*p* = 0.02), respectively. Readmissions for post-TIPS hepatic encephalopathy (HE) in the older and younger groups were *n* = 14 (28%) and *n* = 5 (10%) (*p* = 0.04), respectively. Thirty-day mortality was higher in the older group compared to the younger group, but the difference was not statistically significant, 24% and 12%, respectively (*p* = 0.19). TIPS can be performed safely in patients 70 years and older, but the overall readmissions, and specifically for HE, were significantly higher in older patients. Patients 70 years and older should be followed closely after TIPS, and early introduction of treatment for encephalopathy should be considered.

## 1. Introduction

As the population ages, an increasing number of older patients with cirrhosis are presenting for complications from portal hypertension, manifesting as ascites, hepatic hydrothorax, and variceal bleeding [1]. By 2030, 13% of the population, or 72 million, is estimated to be over age 65 [2]. An estimated 4% of the population has chronic liver disease [3,4]. With the increasing aging population and nonalcoholic fatty liver disease epidemic, a growing number of patients may present with complications from cirrhosis [5].

The most common manifestations of cirrhosis are ascites and varices. Initial medical management involves diet modification, diuretics, serial thoracentesis, paracentesis or nonselective beta blockers and surveillance endoscopy with variceal ligation [6,7]. In certain cases, definitive management cannot be achieved through these methods, resulting in refractory ascites, pleural effusions, or variceal hemorrhage. 

Transjugular intrahepatic portosystemic shunt (TIPS) was developed as a less invasive option than surgical portosystemic shunts to treat ascites and variceal bleeding [8,9]. TIPS is highly effective in preventing recurrent esophageal variceal bleeding, with a re-bleeding rate after TIPS of less than 10%, and it is also highly effective for refractory ascites with response rates exceeding 75% [9,10,11]. However, complications from TIPS include hepatic encephalopathy, stent stenosis, and liver failure [9,10,11,12,13]. Complications after TIPS, specifically hepatic encephalopathy (HE), may be higher in the elderly. 

Prior retrospective studies have evaluated TIPS outcomes in patients 60 years and older [14,15,16] reporting increased rates of encephalopathy or short-term mortality in this older age group [12,14,17,18,19,20,21]. The purpose of our study was to report outcomes after TIPS in patients 70 years of age and older including rates of HE and 30 d readmissions after TIPS. We hypothesized that advanced age alone should not be a contraindication to TIPS, and in select patients who are 70 years and older TIPS can be effective with acceptable complications. 

## 2. Experimental Section

### 2.1. Patient Selection

We performed a retrospective chart review of all decompensated cirrhotic patients who underwent TIPS at our institution between January 2008 through June 2018. Four hundred fifty-one patients were identified in the initial selection of all patients undergoing TIPS for portal hypertension complications. Of these, 51 were over 70 years of age and 169 under the age of 60. Elderly patients were defined as those 70 years and older and were matched with those 50–59 years old by year of TIPS procedure and the Model for End-Stage Liver Disease sodium (MELD-Na) (±3) in a 1:1 ratio. Patients under 60 or over 70 years of age who did not meet the data matching requirements among selection groups were excluded, resulting in a sample size of 50 patients per age group (Figure 1). We chose to match for year of TIPS procedure in order to account for provider variance in procedure techniques and patient selection criteria for TIPS. We chose the control group of 50–59 years old to have an adequate age gap from the elderly population and because it is the most common cohort of patients undergoing the procedure at our institution.

### 2.2. Objectives of the Study 

The primary endpoints of the study were 30 d readmission and 30 d mortality. Secondary outcomes included readmissions for HE within 30 d after TIPS and length of stay. This study was approved by the IRB.

### 2.3. Data Collection

Data were collected on patient demographics, etiology of liver disease and procedure indication, 30 d mortality, 30 d readmissions, pre- and post-procedure HE occurrence, and length of stay (LOS) (Table 1). Laboratory values analyzed included serum creatinine, sodium, albumin, total bilirubin, and international normalized ratio. MELD and MELD-Na data were from the medical record, including grade of HE using the West Haven classification system, where available, and dose of lactulose and/or rifaximin administration.

### 2.4. TIPS Indications

The indications for TIPS were ascites, acute, uncontrolled or recurrent variceal bleeding, and refractory hepatic hydrothorax, which was resistant to conventional medical management. While some patients suffered from more than one portal hypertensive complication, the primary indication was recorded in our data per documentation in the procedure note.

### 2.5. TIPS Technique 

All TIPS procedures were performed with an interventional radiologist at our hospital using standard techniques. TIPS was performed by gaining access to the right internal jugular vein under ultrasound guidance and a wire advanced centrally followed by placement of a vascular sheath. Under fluoroscopic guidance, a 5 French catheter was advanced into the right hepatic vein and over a stiff guide wire, the catheter was exchanged for a balloon occlusion catheter, and wedged hepatic venography was performed to identify the main portal vein branches. The catheter was exchanged for a Colapinto needle (Cook Medical, Indianapolis, Indiana) and passes made to gain access to the portal vein, confirmed by carbon dioxide contrast injection. The parenchymal tract was then predilated with an 8 mm balloon. A long sheath was advanced into the portal vein over which the stent graft was advanced, deployed, and dilated to the desired diameter. The technique for TIPS placement was identical in subjects regardless of age. The portosystemic gradient (PSG) was computed from pressure measurements taken from the right hepatic vein and the wedged hepatic vein pressure.

### 2.6. Statistics 

Means were compared using Student’s t-test for normally distributed variables and Wilcoxon rank sum test for non-normally distributed variables. Fisher’s exact test was used to compare proportions. Multivariable logistic regression was used to control for potentially confounding factors. A *p* value ≤ 0.05 was considered statistically significant.

## 3. Results

Baseline characteristics of the younger (50–59 years old *n* = 50) and older patients (≥ 70 years old *n* = 50) are shown in Table 1. Median ages for the older and younger group were 73 years old and 55 years old, respectively. The most common etiology of liver disease in older and younger groups were nonalcoholic fatty liver disease (NAFLD) and hepatitis C, respectively. Older subjects had higher serum creatinine compared to the younger group, median creatinine 1.3 and 1.1 mg/dL, respectively (*p* = 0.06). Most patients were Child–Pugh–Turcotte class B patients in both groups of patients, 38% and 40%, respectively. Prior to the TIPS procedure, treatment for HE was with lactulose with or without rifaximin in 15 (30%) younger patients and 11 (22%) older patients (*p* = 0.49). There were no significant differences in other variables between the older and younger groups (Table 1). 

The most common indication for TIPS in nonelderly and elderly was variceal bleed (28, 56%) and ascites (26, 52%; *p* = 0.07) respectively (Table 2). Of the 28 younger patients who underwent TIPS due to variceal bleeding, 17 (61%) patients underwent TIPS emergently for an acute variceal bleed, while 11 (39%) underwent TIPS electively for recurrent episodes of variceal bleeding with band ligation. Twenty-two older patients underwent TIPS for variceal bleed (44%; *p* = 0.3), of which 16 (73%) required TIPS urgently. Pre- and post-TIPS hepatic venous pressure gradients were documented and were not significantly different between the two groups (Table 2).

Table 3 reveals outcomes after TIPS between the two groups. The median (IQR) length of stay for the younger and older groups was 1 (3) and 2 (2) d respectively. Post TIPS, more younger subjects were prescribed lactulose and rifaximin than the elderly subjects, *n* = 43 (86%) and *n* = 38 (36%) respectively (*p* = 0.31). Thirty-day readmissions were higher in the older group compared to the younger group, *n* = 17 (34%) and *n* = 6 (12%), respectively (*p* = 0.02). Readmissions for post-TIPS HE in the older and younger groups were *n* = 14 (28%) and *n* = 5 (10%), respectively *p* = 0.04. There was higher 30 d mortality in the older group compared to the younger group, but the difference was not statistically significant, 24% and 12% respectively, *p* = 0.19.

Characteristics of those who died within 30 d of TIPS are shown in Table 4. All 6 younger patient deaths were due to multisystem organ failure (MOF) within the initial TIPS hospitalization. Five of these patients (29%) had TIPS performed due to acute variceal bleeding with baseline MELD scores ranging from 12–27. One (6%) had TIPS performed for refractory ascites, with a MELD score of 17 and died due to sepsis from bacteremia and MOF. In the younger group, 12 older patients died within 30 d of TIPS placement. Six (38%) died who underwent TIPS for an acute variceal bleed. Five of these patients died due to MOF within the initial TIPS hospital stay (range 0–9 d), and one patient went home with home hospice, dying 18 d later. Among the older who had TIPS placed for ascites, 5 (20%) died within 30 d of TIPS placement (range 15–28 d). Baseline MELD scores of the 12 patients ranged from 10–27, of which 6 had a MELD score of 18 or greater. There were 5 patients over the age of 80 in our chart review, 3 of which presented for ascites management and died within 30 d of TIPS. Of the total 18 patients who died in the elderly and nonelderly groups, 9 had baseline MELD scores less than 18. Of these lower MELD patients, 4 had TIPS placement for acute variceal bleeding, and 4 were placed for ascites with age ranges 77–84. 

Results from the multivariable analysis are shown in Table 5. After adjusting for etiology of liver disease, pre-TIPS encephalopathy, and indication, age 70 years old was associated with a 5.3-fold increased risk of re-hospitalization after TIPS (OR = 5.3, 95% CI 1.4-20.1), *p* = 0.014.

## 4. Discussion

TIPS has revolutionized the way we approach variceal bleeding and ascites. TIPS results in lower re-bleeding rates and improved survival in patients who undergo TIPS for recurrent esophageal variceal bleeding or within 72 h of esophageal variceal bleeding [7,10,11]. The benefit for TIPS and refractory ascites is improvement in quality of life with decreased need for large volume paracentesis, while a survival benefit for this indication has not been consistently demonstrated [15,18,22,23,24,25,26]. A well-recognized complication of TIPS is HE, which can be disabling and more disabling than the indication for TIPS, such as ascites. Older patients and patients with pre-existing HE are at increased risk for refractory encephalopathy after TIPS [12,19,27,28,29].

Most studies that have specifically evaluated TIPS outcomes in older populations have defined elderly to be 60 [16] or 65 [14,15] years and older or have not reported rates of post-TIPS encephalopathy or readmission rates. These studies had total sample sizes ranging from 23-64 patients. The number of patients over 70 years old was small (less than 10) and was likely an influencing factor in choosing the definition of “elderly”. Saad et al. evaluated 90 d survival in 65 cirrhotic patients over the age of 70 in comparison to a much larger cohort of 24–69 year olds. They found lower 90 d survival rates in the elderly cohort (60% vs. 85%), but they did not evaluate HE or readmissions in their study [27]. As the population ages, an increasing number of patients will be in their eighth and ninth decades. Therefore, data are needed in these age groups to adequately determine if TIPS is safe, to provide patients with information on complication rates, and to guide clinicians on optimal management, such as early recognition and treatment of HE.

This is the largest known study in TIPS patients 70 years of age and older directly comparing results to a MELD-Na matched group in a 1:1 fashion. In the elderly, overall readmissions and readmissions specifically for HE were significantly higher than that of the nonelderly, suggesting older patients should be followed more closely after TIPS, and early introduction of treatment for HE should be considered.

There was also a trend towards increased 30 d mortality in the elderly population undergoing TIPS, although it was not statistically significant. The study may have been underpowered to detect a statistically significant difference. The oldest subjects in our group ranging from ages 80–84 years had TIPS placed for ascites. All three died within 30 d post TIPS despite a low baseline MELD ranging from 10–16, suggesting that age over 80 may be associated with a higher mortality than seen with patients in their seventies. [12,14,17,18,19,20,21,27]. We had too few patients 80 years and older to study as a separate group. 

Our study also highlights the high mortality after TIPS for acute variceal bleeding. In the nonelderly and elderly groups, 29% (5/17) and 38% (6/16) died within 30 d after undergoing TIPS placement for acute variceal bleeding, respectively. Four of these patients had baseline MELD scores less than 18. No patients who underwent elective TIPS for variceal bleeding died within 30 d. When discussing TIPS with patients 70 years of age and older, the discussion should focus on the high mortality after TIPS for acute variceal bleeding. In addition, three patients whose MELD scores were 20 or greater who had TIPS performed for uncontrolled variceal bleeding all died within 30 d, suggesting in some circumstances TIPS is futile.

There are several limitations to our study. This is a single-center, retrospective, nonrandomized study and is associated with the caveats of retrospective designs. For example, the degree of encephalopathy was not routinely reported in every note and could not be standardized and reported according to the West Haven classification. HE was more often recorded as present or absent, and/or if the patient was on treatment for encephalopathy. TIPS candidacy in older patients was not standardized and was at the discretion of the hepatologist. This could lead to variability in selection criteria among hepatologists. Ideally, it would be useful to develop a standardized inclusion and exclusion criteria for TIPS candidacy that are specific to older patients. Our comparison group was younger patients who underwent TIPS, but another comparison group would be patients 70 years and older with indications for TIPS but who did not undergo TIPS.

Data on post-TIPS complications are limited in individuals 70 years of age and older, and our study provides useful information about morbidity and mortality post TIPS in this group. Our study suggests readmissions and severe HE requiring hospital admission are significantly higher in patients 70 years and older. This information may be helpful for post-TIPS follow-up and early institution of therapy for HE. TIPS for nonacute variceal bleeding, refractory ascites, and hepatic hydrothorax has reasonable outcomes with acceptable morbidity and mortality, but there is a high mortality when TIPS is placed for acute variceal hemorrhage.

## Figures and Tables

**Figure 1 jcm-09-00381-f001:**
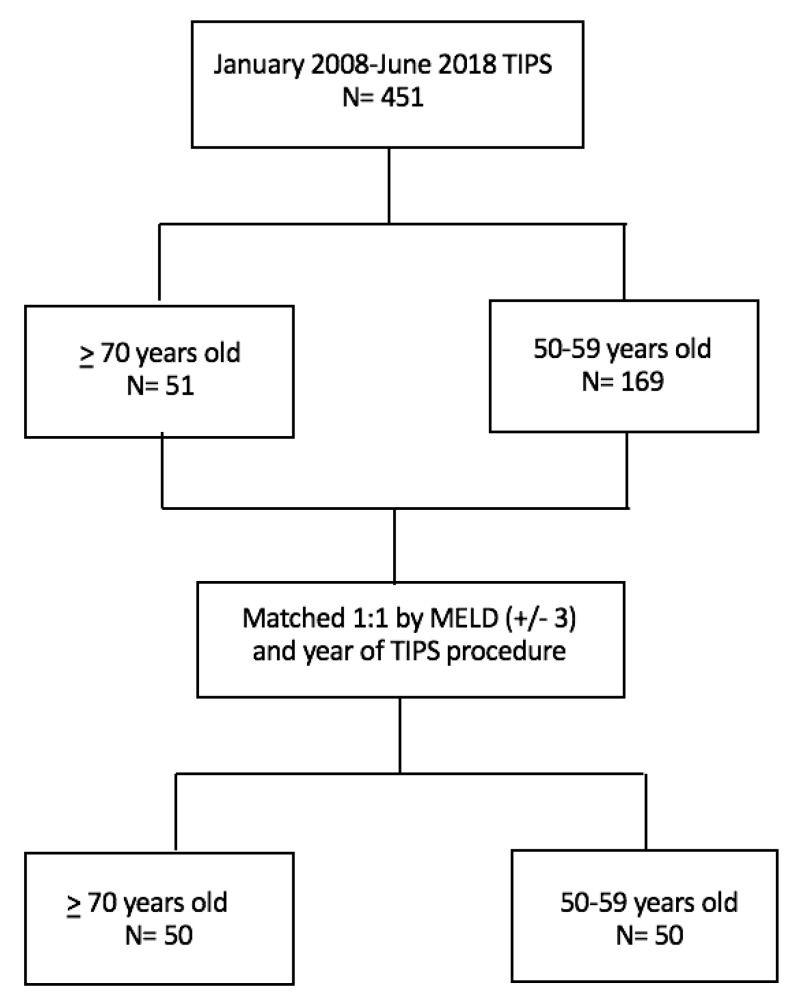
Patient selection.

**Table 1 jcm-09-00381-t001:** Description of study population.

	**Total *n* = 100**	**Age 50–59**	**Age ≥ 70–84**
		*n* (%)	
Age, median (IQR)	65 (18)	55 (4)	73 (6)
Female	40 (40)	18 (36)	22 (44)
White	89 (89)	43 (86)	46 (92)
Black	7 (7)	6 (12)	1 (2)
Asian	2 (2)	0	2 (2)
Other	2 (2)	1 (2)	1 (1)
Hispanic	1 (1)	1 (2)	0

**Etiology of liver disease**		***n* (%)**	
Alcohol	21 (21)	13 (26)	8 (16)
Nonalcoholic fatty liver disease (NAFLD)	33 (33)	5 (10)	28 (56)
Hepatitis C	27 (27)	26 (52)	2 (1)
Autoimmune	3 (3)	1 (2)	2 (4)
Cryptogenic	12 (12)	3 (6)	9 (18)
	**Value (Median IQR)**		
Median MELD-Na	12 (7)	12 (7)	12 (7)
Median MELD	11 (5)	11 (6)	11 (5)
Total bilirubin mg/dL	1.3 (0.8)	1.3 (1)	1.3 (0.8)
INR	1.2 (0.2)	1.3 (1.2)	1.2 (1.1)
Creatinine mg/dL	1.1 (0.5)	1.1 (0.7)	1.3 (0.7)
Sodium mmol/L	138 (9)	138 (6)	138 (6)
Albumin mg/dL	2.8 (1)	2.8 (1.1)	2.9 (1)
**Child–Pugh–Turcotte**		***n***	
A/B/C	12/78/10	4/40/6	8/38/4
On hepatic encephalopathy (HE) treatment pre transjugular intrahepatic portosystemic shunt (TIPS) (lactulose +/- rifaximin)	26 (26)	15 (30)	11 (22)

**Table 2 jcm-09-00381-t002:** TIPS data in the study groups.

	Age 50–59	Age70–84
**Indication *N* (%)**		
Ascites	16 (32)	26 (52) *p* = 0.07
Variceal bleed	28 (56)	22 (44) *p* = 0.3
Acute variceal bleed	17 (34)	16 (32)
Hepatic hydrothorax	6 (12)	2 (4)
Pre-TIPS gradient mmHg	17 (7)	16 (7)
Post-TIPS gradient mmHg	5 (5)	5 (3)

**Table 3 jcm-09-00381-t003:** Outcomes after TIPS in study groups.

	Age 50–59	Age 70–84
30 d mortality *N* (%)	6 (12)	12 (24) *p* = 0.19
Length of stay (mean/median/IQR d)	3/1/3	3/2/2
Readmission *N* (%)	6 (12)	17 (34) *p* = 0.02
Readmission for HE post TIPS *N* (%)	5 (10)	14 (28) *p* = 0.04
On lactulose post TIPS *N* (%)	43 (86)	38 (76)
On rifaximin post TIPS *N* (%)	18 (36)	22 (44)

**Table 4 jcm-09-00381-t004:** Characteristics of patients that died within 30 d of TIPS.

Age	MELD pre TIPS	Indication	Cause of Death	Days to Death
50	27	Acute variceal bleed	Multisystem organ failure (MOF), liver failure	0
52	17	Acute variceal bleed	MOF, liver failure	1
53	14	Acute variceal bleed	MOF, liver failure	3
55	14	Acute variceal bleed	MOF, liver failure	8
58	12	Acute variceal bleed	MOF, liver failure	1
56	17	Ascites	Sepsis, MOF, liver failure	22
70	18	Acute variceal bleed	Liver failure, hospice	18
70	24	Acute variceal bleed	MOF, liver failure	1
71	11	Ascites	HE, fell broke hip, home hospice	25
72	27	Acute variceal bleed	MOF, liver failure	1
72	18	Acute variceal bleed	In hospital	3
74	10	Acute variceal bleed	MOF, liver failure	9
75	20	Hepatic hydrothorax	MOF, liver failure	9
77	11	Ascites	MOF, liver failure	15
79	18	Acute variceal bleed	MOF, liver failure	0
80	10	Ascites	Home hospice, liver failure	28
83	10	Ascites	Unknown	17
84	16	Ascites	UTI, sepsis	23

**Table 5 jcm-09-00381-t005:** Results of multivariable analysis for readmission.

**Variable**	**OR (95% CI)**
Age ≥ 70 years old	5.3 (1.4–20.1)
Pre-TIPS encephalopathy	1.4 (0.5–4.3)
Ascites as indication	1.6 (0.3–10.5)
Variceal bleeding as indication	1.2 (0.2–8.0)
Etiology: Alcohol	1.4 (0.3–5.9)
NAFLD	1.2 (0.4–4.2)
Viral hepatitis	3.5 (0.7–17.1)

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
