# Peer review of "Outcomes After Transjugular Intrahepatic Portosystemic Shunt in Cirrhotic Patients 70 Years and Older"

_jcm, 2020, doi:10.3390/jcm9020381_

Round 1

Reviewer 1 Report

In patients with baseline Hepatic encephalopathy, do you have the degree of their HE (west haven classification). In Table 2 - define the difference between "variceal bleed" and "acute variceal bleed" Why not to compare patients with an indication emergent vs non emergentIn Table 4. usually MELD predicts mortality in patients undergoing TIPS - almost half of the patients that died had a low MELD. Any comments on this. As we get more elderly people with cirrhosis and complications of liver disease its important to comment on this. Excellent manuscript

Author Response

Thank you for your comments and suggestions. Please find our responses below: 

Point 1. "In patients with baseline Hepatic encephalopathy, do you have the degree of their HE (west haven classification)."

Response 1: We appreciate the reviewer's comments. Degree of encephalopathy was not routinely reported in the notes. It was recorded as present or absent, and/or if the patient was on treatment for encephalopathy (lactulose/rifaximin). We have noted this on page 3 of our study and we recognize this as a limitation in our discussion on page 8. 

Point 2: "In Table 2 - define the difference between "variceal bleed" and "acute variceal bleed"

Response 2: We have clarified this on page 4 in the body of the text to describe the difference between acute variceal bleed (emergent/uncontrolled) and "variceal bleed" (presenting electively for TIPS due to recurrent variceal bleed episodes).  

Point 3:  "Why not to compare patients with an indication emergent vs non emergent in Table 4." 

Response 3: We appreciate the reviewer's comments. Acute variceal bleeding has now been delineated and results for emergent nonemergent are shown in table 4 on page 6, indicating an emergent life-saving procedure. All other indications for TIPS were done on a non-emergent basis. In review of the non-emergent cases, it was noted that one patient was mislabeled as "ascites" indication for TIPS and this was corrected to "acute variceal hemorrhage." This did not significantly impact any results or discussion points of the manuscript.  

Point 4: "Usually MELD predicts mortality in patients undergoing TIPS - almost half of the patients that died had a low MELD. Any comments on this. As we get more elderly people with cirrhosis and complications of liver disease its important to comment on this."

Response 4: We appreciate the reviewer's observation and comments. We have added results on this on page 6 of the text. Those who died with low MELD were older (77-84 years old) and/or indication for TIPS was emergent for acute variceal bleeding, indicating futility of procedure despite lower MELD. Futility of acute variceal bleeding is commented on page 7-8 in the discussion.   

Reviewer 2 Report

Currently TIPS should represent a “bridge” before liver transplantation. In the present work, TIPS is described as a definitive solution to treat cirrhosis complications. Authors should underline the real meaning of the procedure and explain why it should be used in elderly patients. Please explain the acronym HVPG. Tables should be self explanatory: data are not so clear; please explain. 

Author Response

Thank you for your comments and suggestions. Please find our responses below: 

Point 1: "Currently TIPS should represent a “bridge” before liver transplantation. In the present work, TIPS is described as a definitive solution to treat cirrhosis complications. Authors should underline the real meaning of the procedure and explain why it should be used in elderly patients."

Response 1: We appreciate the reviewer's comments and observations. At our center and additional centers in the US, TIPS is considered a destination therapy in those who are not transplant candidates for various factors (age, comorbidities, etc) who suffer from complications of portal hypertension that cannot be adequately medically managed. Although there is no absolute age cut-off for liver transplant in the U.S., in our study, none of the elderly group over 70 years old were liver transplant candidates due to comorbidities or frailty.  

Point 2: "Please explain the acronym HVPG."

Response 2: This has been clarified on page 5 of the manuscript as hepatic venous pressure gradient.  

Point 3: "Tables should be self explanatory: data are not so clear; please explain."

Response 3: We appreciate the reviewer's comments. We have further clarified the data and labeling. Please see the manuscript tables 1-5, on pages 4-7 for updated labeling.